# Live to Die Another Day: Regeneration in *Diopatra aciculata* Knox and Cameron, 1971 (Annelida: Onuphidae) Collected as Bait in Knysna Estuary, South Africa

**DOI:** 10.3390/biology12030483

**Published:** 2023-03-21

**Authors:** Stephanie Schoeman, Carol A. Simon

**Affiliations:** Department of Botany and Zoology, Stellenbosch University, Stellenbosch 7405, South Africa

**Keywords:** bait polychaete species, regeneration, sublethal predation, dispersal, management/conservation

## Abstract

**Simple Summary:**

The estuarine moonshine worm, *Diopatra aciculata*, is used extensively as bait in the Knysna Estuary in South Africa. During collection, the worm frequently breaks into multiple pieces. If discarded or unused pieces can regenerate to form separate individuals, the population may be maintained, or even increase, despite harvesting. This study investigated bait collecting habits of local fishermen and the natural incidence of regeneration in *D. aciculata*. Fishermen usually removed only part of the worm, leaving its tail in the tube and more than half the fishermen return up to 50% of bait collected to the estuary. Naturally occurring *D. aciculata* can regenerate missing anterior and posterior chaetigers, but only if amputation occurs before the 17th or after the 21st segment. Most unused fragments are probably too small to recover from damage inflicted during bait collection, so regeneration is unlikely to cause population expansion despite harvesting. However, some fishermen do move bait from the estuary. Range expansion can therefore occur if large fragments discarded at fishing sites in other estuaries do regenerate, forming new populations.

**Abstract:**

Regeneration is critical for survivorship after injury, sublethal predation, and asexual reproduction; it allows individuals to recover, potentially enabling populations of bait species to overcome the effects of bait collection through incidental asexual reproduction. Opportunities for regeneration are created when worms break during collection (which happens more often than not) and are thrown back into the estuary. Additionally, the trade and movement of bait could result in the range expansion of invasive species. This study investigated bait collection habits of local fishermen and the in situ incidence of regeneration in the estuarine moonshine worm, *Diopatra aciculata*. The evidence shows that this species is capable of anterior and posterior regeneration. The disproportionately small percentage of worms that seem to be recovering from the degree of damage that may be inflicted during bait collection suggests that regeneration may not help worms to withstand the effects of bait collection. However, the continuous movement and discarding of even small numbers of bait in other estuaries can lead to range expansion through incremental build-up, forming new populations, if these fragments are large enough to regenerate.

## 1. Introduction

Globally, members of 12 of the 81 families of polychaetes are used as bait [1,2] with several countries, including Britain [3], the Netherlands [4], Taiwan [5], and Australia [6], commercially producing species for fish feed, bait sales, and bait exportation [1,7]. Among the taxa used globally, onuphids are one of the more widely used [1,8,9,10,11]. More specifically, members of the genus *Diopatra* Audouin and Milne Edwards, 1833, are widely used and highly valued as bait [12,13]. *Diopatra* are dug up in estuaries in Turkey [14] and Italy [6], (*D. neapolitana*), Spain [15], Portugal [16] and France [15,17], (*D. biscayensis* and *D. neapolitana*), and Australia (*D. aciculata*) [6]. *Diopatra aciculata*, which was first described in Australia inhabiting estuaries of New South Wales, Victoria, South Australia, and Western Australia [18], is considered one of the five most expensive bait polychaete species worldwide [13]. The species is currently documented in Australia [19], Egypt [20], and South Africa [21].

In South Africa, the three most widely reported polychaete genera used as bait are *Arenicola*, *Pseudonereis,* and *Gunnarea* with isolated reports of onuphids used as bait ([22,23] Supplementary Material Tables S1–S3). One of these is of *Diopatra cuprea,* used in KwaZulu Natal on the east coast [24], although the collection of polychaetes has subsequently been prohibited in that province [23]. *Diopatra neapolitana* was reported in low densities in Swartkops [25] and Knysna in the 1950s [26], and in Keurbooms in the 1980s [27]. However, *D. neapolitana* or a *Diopatra* sp. was only reported as bait in Swartkops in the 1970s and 1990s [28,29] and in Knysna in 2004 [30]. It was only in 2018 that Van Rensburg et al. (2020) correctly identified the species as *Diopatra aciculata* and categorised the species as cryptogenic, as the records in South Africa [20] may predate the description in Australia [18]. Now known as the estuarine moonshine worm [23], *D. aciculata* is only reported in Knysna, Keurbooms, and Swartkops Estuaries [21] and is the second most popular polychaete bait species in the Knysna Estuary after *Arenicola loveni* [31]. Since the 2000s, the population of *D. aciculata* in Knysna has increased in size with recent population estimates indicating approximately 20 to 24 million individuals [21]. It is clear that the species is undergoing population expansion in Knysna despite extensive utilization as bait [21] suggesting that strategies exist allowing the animals to flourish [31]. A clue to this conundrum may be linked to how worms are harvested—by hooking it out of its tube ([20,29] Supplementary Material Video S1) which usually breaks the worms at various places along the body. The size of the fragment removed depends on the skill of the bait collector and predominantly consists of the anterior portion of the worm including the branchiae. If these fragments which are left behind regenerate, an opportunity for damaged worms to recover and persist despite harvesting is created.

Regeneration plays a key role in survivorship after injury, sublethal predation, and asexual reproduction in many members of Annelida [32,33,34,35,36]. Regeneration abilities, however, vary greatly, ranging from cellular regeneration to being able to regenerate an entire specimen from mid-body chaetigers [37]. This high variability is likely linked to the body plan. Most of the body of annelids is comprised of repeated chaetigers separated from each other by septa with the only non-segmental parts being the head and pygidium [38]. Most individuals contain a fixed number of chaetigers, so if any are lost during injury, a fixed number will be regenerated [39]. Posterior and anterior regeneration involves constriction of the wound and the formation of a blastema containing stem cells, but these processes differ critically [40]. During posterior regeneration, the first step after blastema formation is the formation of a new pygidium with a functioning anus [40]. Thereafter, chaetigers are added individually as normal in the posterior growth zone in front of the pygidium until all lost chaetigers are replaced [40,41]. During anterior regeneration, the prostomium and all the chaetigers lost are generated at more or less the same time [40,42]. Because the same process is seen during architomy, regeneration after injury can be exploited to create opportunities for incidental asexual reproduction, as seen in the propagation of sabellid species for aquaculture [43]. The ability to reproduce asexually through regeneration, be it intentional or incidental, can have serious management implications as it may facilitate the population and range expansion of species, including non-indigenous species [43].

Regeneration has been observed in 11 species of *Diopatra*, but they differ in regenerative capabilities [36]. Several members of the genus, including *Diopatra aciculata*, *D. dentata*, and *D. maculata* have been documented to only regenerate posteriorly while *D. neapolitana*, *D. cuprea*, *D. micrura*, *D. claparedii,* and *D. marocensis* can regenerate both anteriorly and posteriorly [6,8,35,36,44,45].

Regeneration in *D. neapolitana* has been investigated experimentally in Portugal [35]. The study showed that the species can regenerate anteriorly or posteriorly if fewer than half the branchiate chaetigers were removed [36]. The authors therefore concluded that individuals can recover from sublethal predation when small pieces are removed, but are unlikely to recover from injury due to bait collection when 20 or more chaetigers of the animal are routinely removed [36].

Extensive posterior regeneration linked to density-dependent aggression has been reported for *D. aciculata* in farmed populations [6], at a proportion that far exceeds that observed in any other *Diopatra* species [36]. However, whether *D. aciculata* can regenerate anteriorly, and to what extent, is still unknown. *Diopatra aciculata* is morphologically very similar and very closely related to *D. neapolitana* [20] and it is therefore possible that they would show similar regenerative potential. The branchiae in *D. aciculata* start from the fourth or fifth chaetiger and extend for 20 to 40 chaetigers [21], whereas the branchiae in *D. neapolitana* start at chaetiger three or four and extend for approximately 45 to 55 chaetigers [36]. *Diopatra neapolitana* can regenerate anteriorly if the amputation site was around the 15th chaetiger, whereas amputation at the 20th chaetiger led to the death of both parts of the worm. Amputation after the 25th chaetiger, however, allowed for posterior regeneration [36]. If the number of chaetigers where regeneration is possible is proportionate to the total number of branchiate chaetigers, and if these are similar in *D. aciculata*, we can expect anterior regeneration if amputation occurs around the 5th to 13th chaetigers and posterior regeneration if amputation occurs after the 9th to 22nd branchial chaetigers in this species.

Worms are often damaged during bait collection, creating fragments that could regenerate in several ways: only a fraction of a specimen is usually removed from the tube, leaving the posterior fragment in situ [46]; the individual may break into pieces after being removed from its tube [47]; and leftover bait is often discarded by fishermen and bait collectors [21,31,48]. If these fragments can regenerate, collecting and discarding bait has the potential to greatly affect population growth. Discarded pieces can regenerate, allowing populations to be maintained while incidental asexual reproduction can even allow for population growth.

The Knysna Estuary is situated within the Garden Route National Park and is managed by the South African National Parks (SANParks), who consequently conduct regular surveys of fishing and baiting activity in the area [49]. Over the period of January to December 2021, they determined that on average 45 people fish per a day during the week and 74 people fish per a day over weekends. This amounted to a total of 19,954 fishing days for that year [49]. Furthermore, they also made 494 observations of baiting and found that 12% were most likely collecting *Diopatra*. Finally, they report that recreational and subsistence fishers collect worms, while subsistence fishers may also sell worms to recreational fishers, including tourists, and that the bait sold by subsistence fishers are often used by recreational fishers to catch fish in areas outside of Knysna [50]. The ability to regenerate may also facilitate dispersal, but while there is evidence that the movement of bait species for trade has been implicated in the spread of invasive species [51], few studies have considered the effects of bait collection and the intraregional movement of this bait species.

This study investigates (1) the incidence of regeneration in *Diopatra aciculata* and (2) bait collecting behaviour to explore the potential for regeneration to facilitate population maintenance or expansion despite harvesting and if it could enable the dispersal and range expansion within Knysna Estuary and to other estuaries.

## 2. Materials and Methods

The Knysna Estuary on the south coast of South Africa (Figure 1) is an S-shaped estuarine bay approximately 20 km long, with a tidal flow and extensive intertidal flats [52]. The system covers an area of 10 km^2^ at low tide and 16 km^2^ at high tide with water supplied from the Knysna river, several smaller northern and eastern tributaries, and the permanently open mouth [53]. The estuary can be divided into three sections: the marine-dominated and strongly tidal lower estuary or “embayment” from the Western and Eastern Heads to the railway bridge; the marine-dominated middle estuary, from the railway bridge to the road bridge, dominated by warm water with strong salinity gradients; and the typically estuarine upper estuary, upstream of the road bridge, which is influenced by fluvial flow [54].

### 2.1. In Situ Regeneration

From January 2021 to June 2022, approximately forty specimens were collected monthly from Bollard Bay and The Point (Figure 1). Individuals were collected using a 1 m length of piano wire ([20] Supplementary Material Video S1), and taken to the laboratory for further analysis. Each individual was examined for signs of regeneration and classified as showing no regeneration, regenerating anteriorly, regenerating posteriorly or regenerating in both directions (bidirectional regeneration). When regeneration was present, the number of original branchiate chaetigers present, the number of chaetigers regenerating anteriorly (excluding the prostomium and peristomium), and the number of chaetigers regenerating posteriorly were recorded. Since a fixed number of chaetigers will appear simultaneously during anterior regeneration [39], the number of chaetigers regenerating can inform the total number of branchiate chaetigers that were originally present and the extent of the damage being repaired. During posterior regeneration, however, chaetigers are added individually after a new posterior growth zone is established [40]; consequently branchiate chaetigers are only replaced near the completion of regeneration. The number of original branchiate chaetigers present therefore cannot be used to determine the exact number of chaetigers lost posteriorly but can be used to estimate the extent of damage to the individuals. Images of regeneration were taken on a Leica Stereomicroscope (Leica microsystems, Wetzlar, Germany; model number: Leica mz7.5) fitted with a Leica microscope camera (Leica microsystems, Wetzlar, Germany; model number: Leica EC3) and an Olympus Targus 5.

### 2.2. Interviews

Bait collectors and fishermen were interviewed throughout the estuary. The interviews were conducted from approximately two hours before low tide to two hours after low tide between June and December 2021. Interviews started at the upper reaches of the estuary, at the road bridge, and concluded near the estuary mouth, at Bollard Bay (Figure 1). Bait collectors and fishermen were identified, approached, and invited to complete the questionnaire. Verbal consent was requested before questioning commenced (human ethical clearance number: REC-2021-19365). The questionnaire was designed to determine the bait preferences, collection practices and post-collection habits of the local fishermen as set out below (Figure 2; Appendix A):To identify the fishermen who use *D. aciculata*, fishermen were asked to list their preferred bait species; only responses from those that selected *D. aciculata* were retained for analysis (Figure 2, Q1).To estimate the magnitude of potential for the regeneration of *D. aciculata*, fishermen were asked how many worms they collected (Figure 2, Q2).To assess if regeneration could lead to dispersal, respondents were asked if they moved bait within and out of the Knysna Estuary (Figure 2, Q3a), as this creates an opportunity for anthropogenic dispersal. Secondly, respondents were asked if they bought *Diopatra* (Figure 2, Q3b), because recreational fishermen tend to purchase bait from subsistence fishermen. As recreational fishermen tend to fish in areas away from the subsistence fisherman (i.e., from whom bait is purchased), the likelihood of anthropogenic dispersal also increases if *Diopatra* are purchased as bait [49]. Furthermore, many recreational fishermen fish outside of Knysna [49].To assess the extent to which discarding unused bait could contribute to dispersal and to maintaining population size despite harvesting, fishermen were asked if they had bait left over and if yes, how the leftover bait was processed or discarded (Figure 2, Q4). In the latter instance, we only considered the discarding of fresh, unprocessed, bait. If large enough pieces of *Diopatra* are thrown back (size gleaned from observational data), a potential for regeneration is created. Once the worm fragments settle and regenerate fully, naturalisation is possible.The fishermen were asked which portion of the worm they preferred as bait (head, middle, tail, or whole worm), together with the frequency with which *D. aciculata* broke during collection (never 0%, rarely 0–33%, sometimes 33–66%, usually 66–99%, always 100%) (Figure 2, Q5a and 5b). This information was used in conjunction with observations of in situ regeneration to explore if broken pieces of worm that are left behind during bait collection could regenerate and contribute to population growth or maintenance. The assumption was that if fishermen predominantly collected the portion of the worm that they preferred to use as bait, this section would predominantly be leftover and discarded, and these sections would therefore have the greatest potential to survive and, if large enough, regenerate. Additionally, the section of the worm left in the tube (i.e., usually the posterior) could also regenerate if large enough. If both anterior and posterior regeneration is possible, both portions can regenerate leading to incidental asexual reproduction.

Responses to the questionnaire were used in conjunction with observations of regeneration and data supplied by SANParks to explore if bait collection and regeneration can facilitate the persistence and anthropogenic dispersal of the species, as set out in Figure 2.

### 2.3. Statistical Analysis

#### 2.3.1. In Situ Regeneration

To calculate the proportion of branchiae that need to be intact for anterior regeneration, the following equation was used:(1)$$\%\;branchiae\;intact=\frac{n(origional\;branchiate\;cheatigers)}{n\left(regenerating\;branchiate\;cheatigers\right)+n(original\;branchiate\;chaetigers)}\times 100$$

The percentage of original branchiate chaetigers needed for anterior regeneration was divided into 10 chaetiger increments (50–59%, 60–69%, 70–79%, 80–89%, and 90–99%). To test whether certain sized fragments were present more frequently than others, a one-way Chi-squared test was performed.

#### 2.3.2. Interviews

To test whether there is a difference in the number of fishermen buying or collecting bait (Figure 2, Q3b), whether leftover bait is discarded more often than not (Figure 2, Q4), and whether bait is moved within and out of the estuary more often than not (Figure 2, Q3a), a one-way Chi-squared test was performed. All analyses were conducted in R Studio (version 4.2.1).

Data obtained from the survey together with the data from SANParks were used to estimate the potential scale of the problem. The number of bait collectors using *Diopatra* in Knysna Estuary was estimated using (2). This estimated value was then used to estimate the number of *Diopatra* caught per year using (3). The number of worms estimated to be extracted annually was used to determine the portion of *Diopatra* discarded per year using (4). The number collected per year and the reported frequency of breaking during collection was used to estimate the total number of potential breakages that can occur per year using (5). Lastly, (6) allows for the estimation of the portion of the worms that are large enough to regenerate and was based on the observations of in situ regeneration.
(2)n(bait collectors using Diopatra)=% prefering Diopatra×n(annual fishing effort from SANParks)
(3)$$n(Dipatra\;caught\;per\;year)=n(mode\;number\;caught\;per\;person\;per\;day)\times n\left(bait\;collectors\;using\;Diopatra\right)$$
(4)n(discarded per year)=n(caught per year)×% discarded from interview
(5)$$n\left(potential\;breakages\right)=n\left(caught\;per\;year\right)\times frequency\;of\;breakage\;from\;interview$$
(6)$$n\left(capable\;of\;regeneration\right)=n\left(discarded\;per\;year\right)\times \%\;that\;show\;regeneration\;from\;in\;situ\;observation$$

## 3. Results

### 3.1. In Situ Regeneration

From January 2021 to June 2022, a total of 594 specimens were collected, with 54.88% (*n* = 326) showing signs of regeneration. There was no difference in the incidence of regeneration between the two chosen sites. Of the 326 regenerating worms, 95.09% (*n* = 310) showed signs of anterior regeneration only (Figure 3B and Figure 4A,B), 1.23% (*n* = 4) showed signs of posterior regeneration only (Figure 3C), and 3.68% (*n* = 12) showed signs of both anterior and posterior regeneration (Figure 3A). The number of chaetigers regrowing in those regenerating anteriorly ranged between 7 and 17 (median = 10, mode = 9) (Figure 3B and Figure 4A). All individuals regenerating anteriorly had 59–100% of the original branchiae intact. Regeneration was more prevalent in the individuals that had a higher percentage of original branchiate chaetigers intact (χ^2^= 886.04, *n* = 313, df = 5, *p* < 0.001) with most of the individuals falling in the 80–89% category (Figure 4B). Furthermore, most individuals were regenerating eight to 13 chaetigers and had 80–89% of their original branchiae intact (Figure 4B). 

Worms showing posterior regeneration displayed a wider variation in the number of chaetigers regrowing, from as few as eight to as many as 82 (median = 32; mode = 22) (Figure 3C and Figure 5). The individuals that showed signs of both anterior and posterior regeneration had a minimum of 21 and a maximum of 67 (median = 45; mode = 42) original branchiate chaetigers intact (Figure 3A and Figure 5).

### 3.2. Fishermen Baiting Habit Survey

Seventy fishermen and bait collectors were interviewed throughout the Knysna Estuary. Of these, 35 were recreational and 35 were subsistence. Only 23 (32.86%) bait collectors and fishermen (16 recreational and 7 subsistence) selected *D. aciculata* as their preferred bait, and their responses were retained for further analysis. The respondents indicated that they collected a minimum of 1 and a maximum of 96 worms per day (median = 10, mode = 10).

The fishermen who collected *D. aciculata* did not show a statistically significant trend towards moving bait from collection sites within Knysna (χ^2^ = 3.522, *n* = 23, df = 1, *p* = 0.061), although a significantly greater number of fishermen do not move from bait collecting to fishing sits outside of the Knysna Estuary (χ^2^ = 14.727, df = 1, *n* = 23, *p* = 0.0001) (Figure 6A). Significantly more individuals collected their own bait compared to those who bought bait (χ^2^ = 15.696, df = 1, *n* = 23, *p* < 0.0001) (Figure 6B).

A statistically significant proportion of the fishermen (*n* = 20) had bait leftover at the end of a fishing trip (χ^2^ = 12.565, df = 1, *p* = 0.0004), and although more than half the fishermen interviewed indicated that they threw leftover bait back into the estuary or sea, this was not significantly more than the proportion who kept the worms to use on a later fishing trip or donated them to other fishermen (χ^2^ = 4.9, df = 2, *p* = 0.0863) (Figure 6C). Fishermen admitted to throwing away 10% to 50% of the bait they collected each trip (mode = 50% bait discarded per fisherman).

Fishermen noted that worms would break during bait collection significantly more frequently than not (χ^2^ = 15.875, df = 4, *p* = 0.003) (Figure 6D). Most fishermen indicated that worms broke 66-99% of the time during collection. The portion of the worm preferred as bait varied (χ^2^ = 16, df = 3, *p* = 0.001) but most preferred the head or whole worms where possible.

A total of 19,954 bait collection efforts took place over a 12-month period [49] (Table 1). Using the proportion of bait collectors from our survey (32%) and SANParks (12%) ((2) to (6)), we can estimate approximately 23,945 to 63,853 worms are collected annually and 11,972–31,926 are discarded. If only 15% had 60–79% of the original branchiae intact (Figure 4B), 1796 to 4789 individuals of those discarded are big enough to regenerate each year. Additionally, if 63,853 worms are caught per year and worms break 66–99% of the time during collection, pieces from 15,804–63,214 worms are left behind after collection.

## 4. Discussion

This study demonstrates that *Diopatra aciculata* has a great capacity for regeneration. Fifty-four percent of the individuals examined showed signs of anterior regeneration. However, the comparatively few chaetigers (less than 20% of the branchiate chaetigers) lost and being replaced anteriorly may reflect recovery from sublethal predation, rather than bait collection [36]. The natural predators of the genus include fish [29], birds [55], and crustaceans [56]. For example, the spotted grunter (*Pomadasys commersonnii*), a known predator of the species [29], is found in the Knysna Estuary. Additionally, the African sacred ibis (*Threskiornis aethiopicus*) was often observed feeding in the intertidal zones during low tide [46]. It is likely that these species, amongst others, are responsible for sublethal predation on *D. aciculata* in Knysna. On the other hand, only 3.05% of the population exhibited signs of posterior regeneration. A high incidence of posterior regeneration was linked to aggression among neighbouring worms in an aquaculture population when density increased to 2000 worms/m^2^ [6]. This level of intra-specific competition is unlikely in the Knysna population where even at the estimated population numbers of 20–24 million individuals, density never exceeded 52 worms/m^2^ (mean density 3.47 worms/m^2^) [21].

Observations of numbers of branchiate chaetigers regenerating anteriorly, in conjunction with the original, intact, branchiate chaetigers, suggest that *D. aciculata* can have 20 to 70 branchiate chaetigers, depending on the size of the specimen, with the maximum nearly double what has been previously reported (20 to 40; [8]). Thus, if successful regeneration can only occur with at least half the original branchiate chaetigers intact [36], worms can survive the loss of approximately 10 to 35 of their branchiate chaetigers. Both anterior and posterior regeneration is greatly dependent on the presence of the branchiae [37]. The branchiae are extensions of the body wall containing loops of the vascular system that increases the surface area for gas exchange [57]. In tube dwellers such as *Diopatra*, the branchiae are located toward the anterior end where most water flows [57]. Therefore, regeneration would only occur in fragments that include the anterior portion of the worm that bears the majority of the branchiae. This is supported by our observations of anterior regeneration only being present when the amputation was before the 17th chaetiger, which is equivalent to at least 60% of the original branchiae intact (Figure 4). Similarly, during bidirectional and posterior regeneration, the smallest number of original branchiate chaetigers present were 21 and 39, respectively (Figure 5). However, regardless of the total number of branchiate chaetigers present, no anterior regeneration was observed past the 17th chaetiger. This suggests that regeneration is only possible if amputation is before the 17th and after the 21st chaetiger. A similar trend was seen in *D. neapolitana* (15th and 25th chaetiger, respectively) suggesting that regeneration is limited by the specific chaetiger where amputation occurs, rather than the proportion of branchiate chaetigers lost [36].

Although bidirectional regeneration is possible for *D. aciculata*, neither *D. neapolitana* nor *D. aciculata* can incidentally reproduce asexually. In *D. neapolitana,* amputation in the mid-branchial region led to the death of both halves of the worm [35]. Similarly, no anterior regeneration was documented if more than 17 chaetigers were removed. We therefore conclude that as for *D. neapolitana* [36], it is unlikely that *D. aciculata* can withstand the damage inflicted by bait collection.

Only a small proportion (15%) of the regenerating worms collected displayed a level of damage that could be attributed to bait collection. This could be due to the methods used to remove bait, by hooking them out with piano wire inserted into individual tubes [21,30]. This targeted collection of *Diopatra*, may result in few individuals capable of regeneration being left behind inside the tube. Even when the whole worm is removed from the tube, it usually breaks into at least two pieces, leaving the anterior-most portion to burrow into the sand [46,47], if not picked up by the bait collector. Therefore, it is likely that worms showing signs of posterior or bidirectional regeneration had either escaped after removal or been discarded by fishermen.

If the ability to regenerate allows for the maintenance of the population despite bait collection, each individual harvested must leave behind a fragment capable of regeneration. This would negate effects of bait collection, restoring population numbers, and ultimately creating an endless supply of bait (Figure 2). Each fisherman can legally collect ten *Diopatra* worms per day [58] with subsistence fishermen active several times a week and recreational fishermen only on weekends and holidays [31]. Based on the results of the survey and in situ observations of regeneration, we estimated that less than 1% of the total population of *D. aciculata* is collected per year. Additionally, 20 of the 23 respondents had bait left over and discarded up to 50% of the bait collected, and it is therefore estimated that approximately 11,972 individuals are discarded per year. Of the discarded bait, 1765 fragments are big enough to settle and regenerate. However, this is a mere 7.37% of the total number that is collected annually, and unlikely to be enough to allow the population to not only withstand the pressures of bait collection, but to facilitate population expansion. On the whole, fishermen indicated that they complied with the daily allowable catch, but it is possible that they were underreporting their catches. For that reason, this estimated portion may be an underestimation. The small portion of individuals that lost more than 20% of their branchiae (15%) and showed signs of posterior and bidirectional regeneration (1.23% and 3.68%, respectively) further suggest that recovery after collection is unlikely. It is therefore clear that the sexual reproductive strategies of the worm must be robust enough to not only counteract the effects of predation and baiting, but also contribute to expansion. An investigation into the reproductive cycle and frequency of spawning is currently underway [59].

The staff of SANParks [50] report that recreational fishermen within a 100 km radius of Knysna frequently travel to Knysna to buy bait for use in other areas. This creates the risk of anthropogenic dispersal [43] if leftover bait that can survive and regenerate is discarded at the fishing site (Figure 2). Only two fishermen admitted to buying and moving bait from Knysna, but this is probably an underestimation. Selling and by extension buying live worms is illegal [58], and interviewees may avoid incriminating themselves or guilty fishermen were not interviewed (see also [31]). Furthermore, the high unemployment rate in the area [60] is leading to an increase in subsistence fishermen and bait collectors illegally selling bait to recreational fishers. Therefore, this number is likely to increase in the future. Nevertheless, even if only a few worms are transported this way, the consistent movement of bait could create a high enough propagule pressure [61], resulting in the development of a self-sustaining population.

## 5. Conclusions

In conclusion, although *Diopatra aciculata* is capable of anterior and posterior regeneration, our data suggests that regeneration will not allow the species to withstand the effects of bait collection. However, the consistent movement of bait to other estuaries by fishermen can, under the right circumstances, lead to the development of new populations of the worm and the anthropogenically aided dispersal of the species. However, further research that investigates the dispersal capabilities of the species is required to strengthen this conclusion.

## Figures and Tables

**Figure 1 biology-12-00483-f001:**
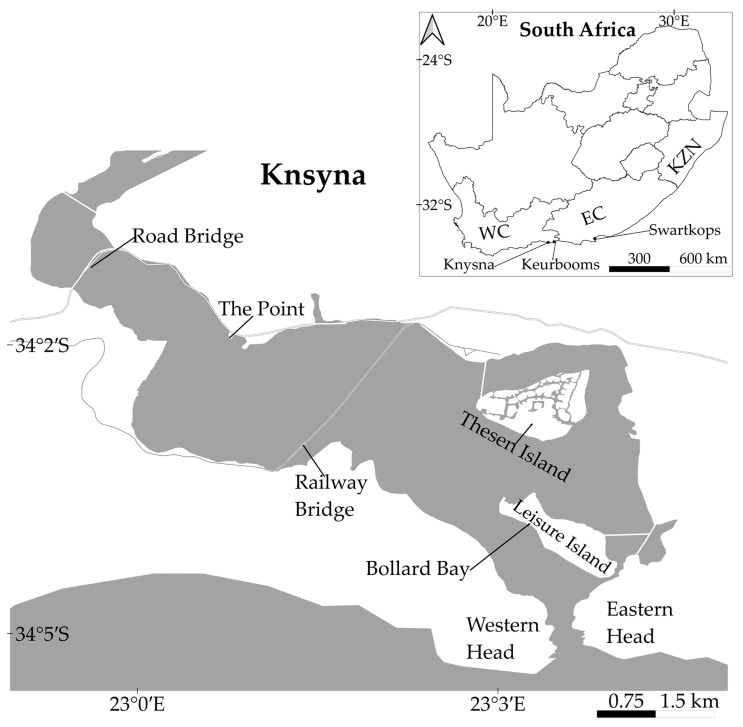
Knysna Estuary on the south coast of South Africa, showing the sampling sites (The Point and Bollard Bay). Key: WC: Western Cape; EC: Eastern Cape; KZN: KwaZulu-Natal.

**Figure 2 biology-12-00483-f002:**
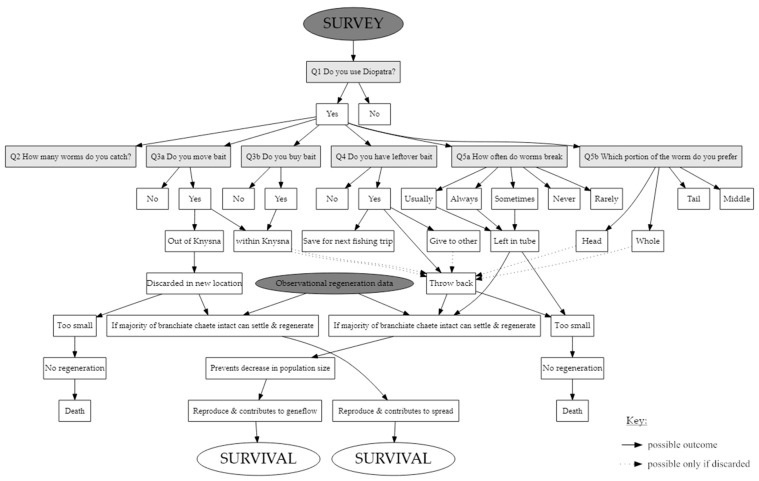
A flow diagram depicting how information from the in situ regeneration and survey were used to assess if bait collection and regeneration can contribute to anthropogenic dispersal.

**Figure 3 biology-12-00483-f003:**
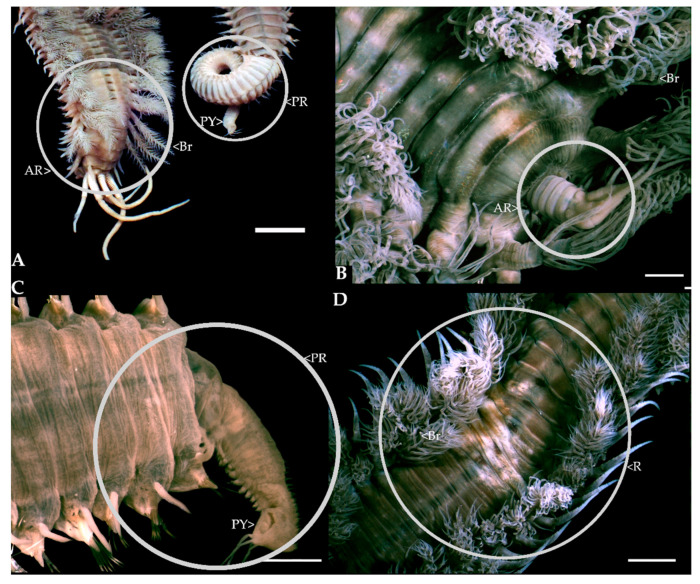
Evidence of regeneration in *Diopatra aciculata* in the Knysna Estuary: (**A**) bidirectional regeneration, (**B**) anterior regeneration; (**C**) posterior regeneration; (**D**) regeneration of chaetigers in the branchial region; Key: R: regeneration; Br: branchiae; PY: pygidium; PR: posterior regeneration; AR: anterior regeneration (scale bar: (**A**) = 0.5 cm (**B**–**D**) = 2 mm).

**Figure 4 biology-12-00483-f004:**
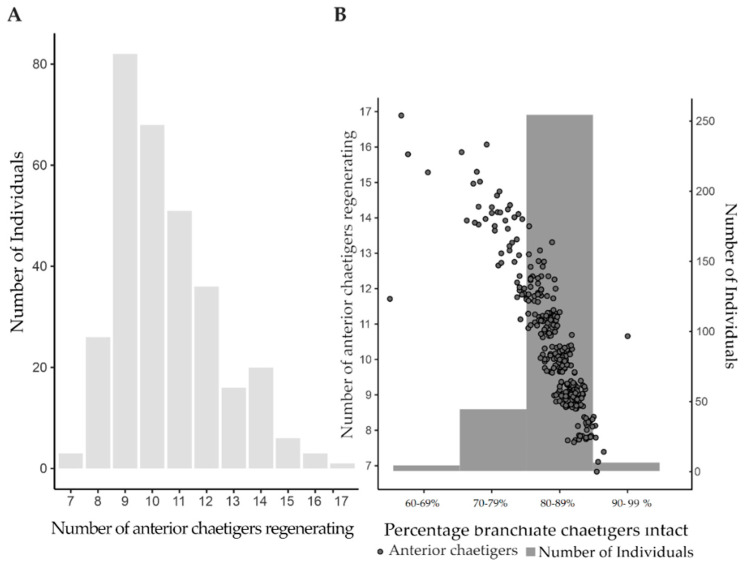
Incidence of anterior regeneration by posterior fragments. (**A**) The number of documented anterior chaetigers regenerating for each individual; (**B**) percentage of the original branchiate chaetigers intact during regeneration and the corresponding number of anterior chaetigers regenerating.

**Figure 5 biology-12-00483-f005:**
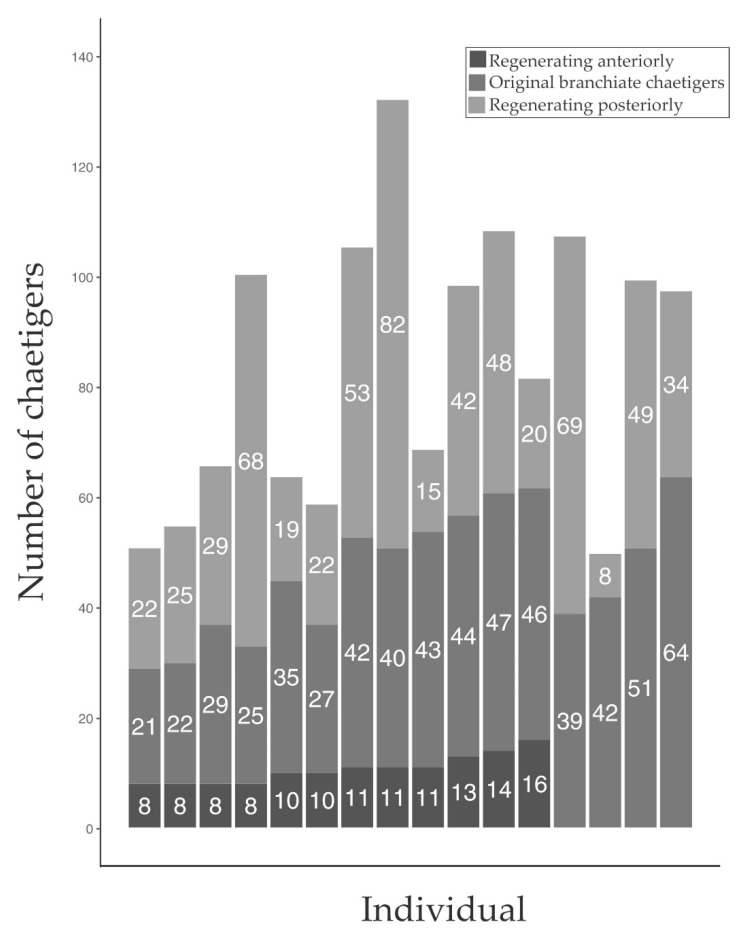
The number of branchiate chaetigers present and the number of chaetigers regenerating in those individuals that showed bidirectional and posterior regeneration. The numbers on the bars represent the number of chaetigers in each category.

**Figure 6 biology-12-00483-f006:**
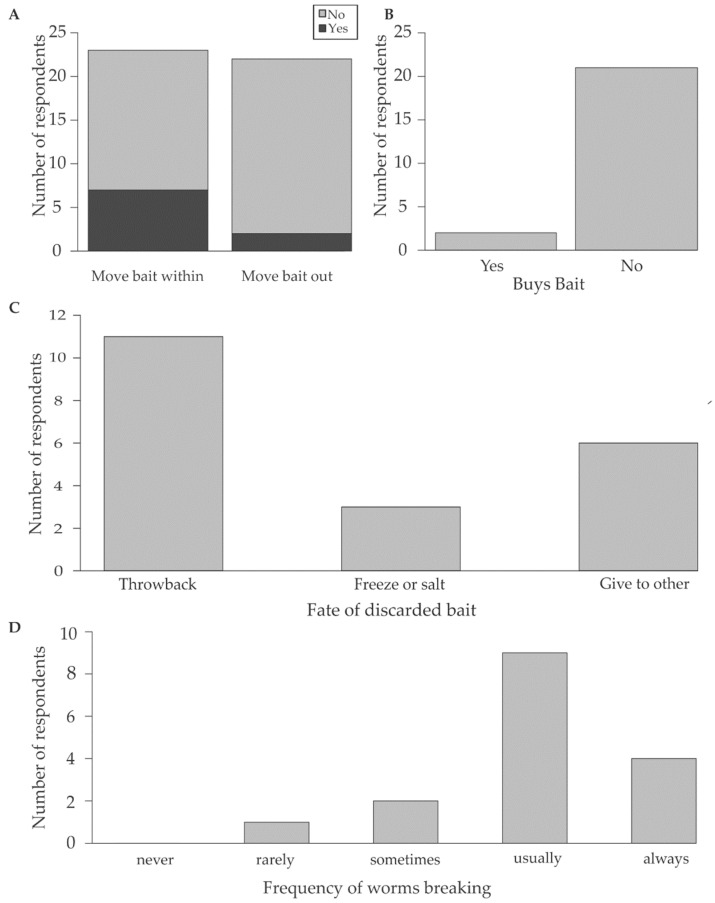
Responses to survey questions: (**A**) Is bait moved within and out of the estuary? (**B**) Do you buy bait? (**C**) What happens to leftover bait? (**D**) How often do worms break during collection?

**Table 1 biology-12-00483-t001:** A breakdown of the number of fishermen collecting *Diopatra aciculata* and the approximate number they caught and discarded per year.

Percentage Preferring *Diopatra*	Number of Bait Collection Opportunities per Year	Number Using *Diopatra* (2)	Number Caught per Year (3)	Number Discarded per Year (4)	Number of Worms Breaking per Year (5)	Number Capable of Regeneration (6)
12% *	19,954	2394	23,945	11,972	15,804–23,705	1796
32% **	19,954	6385	63,853	31,926	42,143–63,214,	4789

*: Taken from the SANParks data. **: Obtained from the survey.

## Data Availability

Not applicable.

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
