# Peer review of "Live to Die Another Day: Regeneration in Diopatra aciculata Knox and Cameron, 1971 (Annelida: Onuphidae) Collected as Bait in Knysna Estuary, South Africa"

_biology, 2023, doi:10.3390/biology12030483_

Round 1

Reviewer 1 Report

Line 25            change is to are

Line 45            ITIS has Diopatra cuprea cuprea listed as unverified, Italian worms are likely D. neapolitana

Line 55            ‘was’ not ‘Was’

Line 115          so your designation of the 15th chaetiger here does not appear to be based on percentage removal or numerical removal. Add a few words to clarify. Note Pires et al. 2012 did not find that the ability of a fragment to regenerate was based on the percentage remaining but rather that the ability to regenerate was based on actual location on the worm.  Given that finding which is consistent with other polychaetes such as the maldanids, I am surprised by your decision here to invoke a percentage.

Line 133          ‘fishers that are’ remove ‘that’

Line 230          ‘original’ not ‘origional’

Line 230          ‘chaetigers’ not ‘cheatigers’

Line 360          I am confused by this calculation.  First you tell me that regeneration only occurs with 50% of the branchiate chaetigers intact (earlier you stated that they had 20 to 40 branchiate chaetigers), so 50% would be up to 20 branchiate chaetigers not the 35 you state here as possibly lost.  Your value of 35 is of course also in conflict with the data presented in Fig. 4 where the highest value you have for regenerating number of anterior chaetigers is 17 of which presumably only 12 to 13 are branchiate.

My suggestion would be that you start this paragraph with several sentences about the revision to the number of branchiate chaetigers i.e. from 40 to 70 and give data on the distribution of those values so that one understands the likelihood of an individual with more than 40.  Then tell us how many they supposedly could lose if 50% was the correct value.  You do then need to use the actual data in Figure 4 which suggests strongly that either individuals with more than 40 or even 35 branchiate chaetigers are rare or that the ability to regenerate is not a function of the percentage of branchiate chaetigers as you suggest but rather the presence of a particular chaetiger as Pires et al. 2012 suggested for Diopatra neapolitana—anteriors regenerated posteriors only if 15 chaetigers or more, posteriors regenerated anteriors only if amputation occurred prior to chaetiger 20.  I think you need to entertain this as a possibility. The end of this paragraph implies this but you need to be more forthright about this suggestion.

Line 383          use of umbrella wires to remove Diopatra as well as salt addition to tubes is common in Portugal among commercial bait diggers.  I have seen this often in the Aveiro Estuary.  Salt addition is very common in Spain, though of course it is illegal.

Line 412          rewrite ‘despite is plenty’

Line 413          ‘are’ not ‘is’

Reviewer 2 Report

This is an interesting investigation of South African fishermen's bait-gathering habits and in-situ regeneration of the estuary moonshine worm Diopatra aciculata. It was well planned and carefully implemented. The authors used complex data analysis to answer their original questions. It is well written and I read it with great interest.

My only two comments are given below.

(Annelida: Onuphidae) should be added to the title of the manuscript, after the name of the species, to show the systematic position of the object under investigation.

Figure 3: Lines 277-281: The letter B in the upper right corner is hard to see. The clapping of the plate would be more visible if it were separated by while lines. Parts of the plate are marked with uppercase letters, while in the legend they are marked with lowercase letters - in both cases, the same letters should be used.

Reviewer 3 Report

This study aimed to investigate the incidence of regeneration in Diopatra aciculata and bait collecting behaviour to explore the potential for regeneration to facilitate population maintenance or expansion despite harvesting and if it could enable the dispersal and range expansion within Knysna Estuary and to other estuaries.

In general, the manuscript is good and well written, and some points should be clarified. My only main concern is about the number of fishermen and bait collectors were interviewed. 70 seems to be a very low number, regarding the number of the total fishermen observed in a season. However, I understand that usually these people does not want to collaborate, thus I want to believe that the authors made an effort to have the maximum number of interviews that was possible.

Introduction:

The authors stated that: “Diopatra are dug up in estuaries in Turkey [14], Spain, Portugal [15], France (D. neapolitana) [16], Italy (D. cuprea cuprea) [7], and Australia (D. aciculata) [6].” The species name after the country means the species that are usually harvested in these country? For example “France (D. neapolitana)” means that the species that is captured in France is D. neapolitana?. If yes, I disagree. In France it was in described in 2012 a new Diopatra species, D. biscayensis, that, due to its size I also believe that is also captured to be used as bait. Later, this species was also reported in Spain:

Fauchald, K.; Berke, S.K.; Woodin, S.A. (2012). Diopatra (Onuphidae: Polychaeta) from intertidal sediments in southwestern Europe. Zootaxa. 3395: 47-58.

Arias, Andrés; Paxton, Hannelore. (2015). The cryptogenic bait worm Diopatra biscayensis Fauchald et al., 2012 (Annelida: Onuphidae) – Revisiting its history, biology and ecology. 
Estuarine, Coastal and Shelf Science. 163: 22-36

Also, D. neapolitana also exists in Italy, and due to their size I also believe that is catched to be used as bait.

Line 66-67 – the authors stated that usually worms break when are harvested. Usually, what is the size of the organisms caught?

Diopatra aciculata distribution worldwide – according to the introduction, it seems that this species was only detected in Australia, Italy and South Africa… it’s this true?

In the introduction the authors stated that D. aciculata is the second most popular bait species. what other species are caught in this estuary?

Methodology - regarding the capture of the organisms, does this method of capture ensure that the organisms are captured without any damage? The video that the authors mention that  will be available as supplementary material was not available with the PDF of the manuscript to revise. Another question, how is the D. aciculata organisms stored after harvesting? Will they be able to bury themselves, construct a new tube after some hours/days out of water and out of the tube, even if they are of a reasonable size? If they were damaged during capture, and if they aren’t well stored, I think that the probability of the discarded worms survive is very low.

Discussion

lines 380-381. The authors stated that “Pires et al. (2012) indicated that up 20 chaetigers of D. neapolitana are removed by fishermen during collection”, but, I read the paper and I did not retained this idea. In fact, Pires et al. stated that “What concerns bait digging, as usually more than 20 chaetigers are harvested by collectors, our results indicate that the posterior part that remains inside the tube will not be able to regenerate an anterior end”. I suggest to rewrite this section.

Line 397 - the authors stated that each fisherman only can harvest 10 worms per day. That means that only 10 D. aciculata worms can be harvested by them, or in this number it is also included other worms species? Please clarify.

Reviewer 4 Report

Comments  on the peer-reviewed MS by Stephanie Schoeman  and  Carol A. Simon entitled Live to die another day: regeneration in Diopatra aciculata Knox and Cameron, 1971 collected as bait in Knysna Estuary,  South Africa 4

The manuscript is a very interesting and informative study that sits at the intersection of research on the ecology of the marine polychaete Diopatra aciculata and research on the impact of fishing (bait collecting) on worm populations. Thus, the methodology of the authors is a combination of methods of ecology and sociology.

The article is preceded by an informative introduction, which provides a detailed overview of commercially used species of marine annelids, as well as data on the regeneration and asexual expansion of Onuphidae living on the coast of South Africa.

The methods used by the authors to study the phenomenon of regeneration in Diopatra, as well as the structure of the questionnaire proposed to the fishermen, are adequate to the tasks of the study.  The illustrations are informative.

My comment concerns the discussion section. Discussing the role of fishermen in the utilization of a population of Diopatra aciculata, as well as the role of regeneration in maintaining the abundance of the worm population, the authors say nothing about sexual reproduction, which should play a decisive role in maintaining the abundance of the worm population. The authors do not discuss or provide any data on the density of annually settling larvae and juveniles. Similarly, a study of the size  structure of the population could provide a basis for a more accurate assessment of the possible role of the influence of fishing (bait collecting) on the state of worm populations, as well as the possible role of fishermen and fisheries in the distribution of Diopatra aciculate.

Round 2

Reviewer 3 Report

The authors have taken into consideration the comments and tried to answer them. However there are still two issues that they need to consider:

 Line 45: , Spain (D. biscayensis), Portugal [15], France (D. biscayensis) [16].

The authors stated that in Spain and France only D. biscayensis is harvested. However, D. biscayensis and D. neapolitana were reported in Spain (Arias et al., 2015) and in France (Fauchald et al., 2012; Pires et al 2010). Please revise accordingly.

Fauchald, K.; Berke, S.K.; Woodin, S.A. (2012). Diopatra (Onuphidae: Polychaeta) from intertidal sediments in southwestern Europe. Zootaxa. 3395: 47-58.

Arias, Andrés; Paxton, Hannelore. (2015). The cryptogenic bait worm Diopatra biscayensis Fauchald et al., 2012 (Annelida: Onuphidae) – Revisiting its history, biology and ecology. Estuarine, Coastal and Shelf Science. 163: 22-36

 Pires, A., Paxton, H., Quintino, V., Rodrigues, A.M., Diopatra (Annelida: Onuphidae) Diversity In 558 European Waters With the Description of Diopatra micrura, New Species. Zootaxa 2010, 2395, 17. 559 https://doi.org/10.11646/zootaxa.2395.1.2

Lines 379-382 – the aithors have “Pires et al. (2012) indicated that more than 20 chaetigers of D. neapolitana are removed by fishermen during collection. This is much less than estimated for this study (on average individuals caught in this study had 61 branchiate chaetigers)…” in fact, the study of Pires et al. 2010 only says that more than 20 chaetigers are removed, and not only 20 chatigers, thus the affirmation “This is much less than estimated for this study” seems not be correct. In fact, the same study indicates in the introduction that “often the anterior part (10-15 cm) is collected by bait diggers. I already saw organisms collect by digging with a shovel and it was collected organisms with 30 to 70 or even more chaetigers. So, I think that when the authors stated “more than 20 chaetigers of D. neapolitana are removed by fishermen during collection” they wanted to show that the portion that satys in the tube after bait collection is not able to regenerate.
